# The Rationale for Combining Hypofractionated Radiation and Hyperthermia

**DOI:** 10.3390/cancers16233916

**Published:** 2024-11-22

**Authors:** Priyanshu M. Sinha, Charlemagne A. Folefac, Jens Overgaard, Michael R. Horsman

**Affiliations:** Experimental Clinical Oncology-Department of Oncology, Aarhus University Hospital, 8200 Aarhus, Denmark; charlemagne@oncology.au.dk (C.A.F.); jens@oncology.au.dk (J.O.); mike@oncology.au.dk (M.R.H.)

**Keywords:** hypofractionated radiation, stereotactic radiation, heat, hyperthermia, pre-clinical and clinical studies

## Abstract

The hypofractionated radiotherapy of cancer involves the application of a reduced number of larger doses per fraction than used in typical conventional radiation treatments. Tumors can also be treated with hyperthermia (heating at 40–45 °C), and although such heat treatments alone have no relevance as a cancer therapy, pre-clinical and clinical studies indicate that hyperthermia can enhance the effect of hypofractionated radiation. However, additional pre-clinical and large-scale clinical studies are needed to optimize and establish standard treatment protocols, as well as demonstrate the efficacy and safety of this combination. This review addresses some of these issues.

## 1. Introduction

Radiotherapy is second only to surgery in its use as a curative cancer treatment, with estimates that more than 50% of all cancer patients receive radiotherapy as part of their cancer management [1]. The mainstay of curative radiotherapy has been conventional fractionated schedules that involve a large number of daily treatments with relatively low radiation doses (i.e., 30–33 fractions of 1.8–2.0 Gy), the rationale being to limit the radiation damage to late-responding normal tissues while maximizing the level of tumor damage. However, technological advances in image guidance and radiation delivery techniques now allow larger doses to be given to tumors while at the same time sparing the dose to surrounding normal tissues. Consequently, hypofractionated radiation treatments using fewer radiation fractions but at a higher dose per fraction, typically ≥3 Gy [2], are becoming accepted practice for the radiotherapy treatment of certain tumor types. These include tumors in the lungs [3], head, and neck [4]; brain metastases [5]; gastrointestinal tumors [6]; and breast and rectal cancer [7]. This also includes stereotactic body radiation therapy (SBRT; typically, 1–5 fractions of 5–30 Gy), generally considered to be an extreme form of hypofractionated radiotherapy. The target for radiation damage is cellular DNA, in which the radiation induces a variety of damage, especially single- and double-strand breaks, resulting in loss of reproductive ability (Figure 1) [8]. These breaks are produced either directly through ionizations in the DNA itself or indirectly in other cellular molecules, primarily water, as this constitutes some 70% of mammalian cells, which then diffuse far enough to reach and damage the DNA. Oxygen is critical for fixing the damage formed, which ultimately leads to cell death unless the damage can be enzymatically repaired.

If cells are irradiated under low oxygen levels (hypoxia), the damaged DNA can be chemically restored to its original state; thus, 2.5 to 3 times larger radiation doses are required to produce the same level of cell killing as cells exposed to normal physiological oxygen conditions [14]. Hypoxia is now known to be a characteristic feature of virtually all solid tumors that occurs due to the inability of the tumor vascular supply to meet the oxygen demands of the growing tumor mass [18,19]. Tumor cells, like normal cells, require an adequate supply of oxygen and nutrients to grow and develop. Initially, these factors come from the blood supply of the host tissue in which the tumor originally arises. However, the tumor soon outgrows this supply, and further growth and development can only occur if the tumor develops its own blood supply, which it does from the host vessels through the process of angiogenesis [20]. Unfortunately, the development of this tumor neo-vasculature is unable to keep up with a more rapid rate of cellular growth. Consequently, regions that are oxygen-deprived, low in glucose levels and energy, and have elevated levels of lactate, high extracellular acidity, and high interstitial pressure develop within the tumor microenvironment [18,19,21,22]. Cells that exist in these poor microenvironmental conditions can survive but become resistant to certain types of therapy, especially radiation therapy. Both pre-clinical and clinical studies have now shown hypoxia to be a major factor for radiation resistance [14].

When tumors are irradiated, the level of hypoxia actually increases [23]. This is simply the result of the radiation killing the more radiation-sensitive normoxic cell population. However, at longer time intervals, reoxygenation occurs, and the hypoxic fraction decreases [23]. The degree of reoxygenation is dependent on the tumor type, with some tumors showing full recovery to the pre-treatment situation, while others show minor recovery [23]. Several mechanisms have been suggested for reoxygenation, including decreased oxygen consumption in the damaged cells and improvements in perfusion [24,25,26]. However, apart from directly killing tumor cells, radiation is now believed to be capable of damaging tumor vasculature, which has the potential to complicate the hypoxia situation (Figure 1). As early as 1947, a study looking at histological sections of irradiated mouse adenocarcinomas reported that during the first day after irradiating with 20 Gy, the changes seen in vivo were quantitatively similar to those seen in vitro [27]. Yet, after 24 h, additional effects were observed in vivo, accompanied by marked vascular reductions. These effects increased up to four days after irradiation and were still present on day 10. This led to the author proposing that radiation initially caused direct tumor cell killing, but at later times, a larger cytotoxic effect occurred via the induction of vascular damage. Other groups have since supported the suggestion that radiation-induced endothelial cell death as a major component in the radiation response of tumors [26,28,29,30,31,32,33,34]. However, some argue this may not be a universal phenomenon [35,36]. This effect was originally thought to be dependent on high radiation doses (>10 Gy) being used [15,37], but a more recent study reported substantial damage to tumor vasculature even at doses as low as 4 Gy [38]. Such radiation-induced vascular damage would be expected to result in additional tumor cell killing as a consequence of those tumor cells downstream of the damage being starved of necessary oxygen and nutrients. The induction of vascular damage by radiation will also increase the degree of tumor hypoxia [16,39], which would be expected to have a negative influence on repeated radiation treatments, suggesting the need for additional treatments to overcome this potential resistance. One clinically applicable therapeutic approach that can effectively target hypoxia is hyperthermia [17,40].

## 2. Combining Heat with Radiation

Heat has a number of biological effects on cells, including chromosomal aberrations, mitotic dysfunction, cytoskeletal cleavage, changes in membrane transport and metabolism, and protein denaturation [41]. The ability of heat to kill cells is dependent on the heating temperature and time of heating, such that the higher the temperature and the longer the heating period, the greater the degree of cell killing [9,42,43]. Of all the heat-induced cellular biological effects, protein denaturation shows a similar time–temperature-dependent relationship suggesting this to be the most likely rate-limiting step for cell killing (Figure 1) [11]. Interestingly, heat-induced cell killing significantly increases if cells are maintained under conditions of oxygen deprivation and/or low pH (Figure 1) [10,44]. This ability of heat to kill hypoxic cells has also been demonstrated in vivo [45,46]. In vitro studies indicated that generally long periods of hypoxic exposure were necessary for cell killing [10,44]. In tumors, hypoxia is either chronic, resulting from a diffusion limitation of oxygen, or acute, due to transient fluctuations in blood perfusion [14], and in vivo studies suggested that the killing effect of heat was primarily in the chronic population [45,46]. Chronic hypoxic cells in tumors are those most likely to be associated with nutrient-deprived conditions that give rise to heat-sensitive low pH [14].

The majority of pre-clinical studies that have tried to establish the optimal criteria for combining radiation and heat have used single treatments with both agents, yet the results obtained have played a major role in determining the most relevant clinical application when radiation and heat are given as multiple fractionated treatments [17,40,47]. Examples of the type of results obtained for both in vitro and in vivo systems are illustrated in Figure 2.

Although there is considerable variability between the findings in different models, trends have been observed. The consensus opinion is that the best effect of combining radiation and heat occurs when the two treatments are given at the same time or very close together and that this benefit decreases as the time interval between them increases regardless of whether the heat is applied before or after irradiating, with the response eventually reaching a plateau. For in vivo studies, the enhanced effect seen with a simultaneous treatment is similar in both tumors and normal tissues, but differences begin to occur when the time interval increases. In tumors, the plateau that is reached, with a radiation–heat interval of around 4 h, is always greater than that seen for radiation alone. Whereas in normal tissues, some degree of residual effect is observed when heat precedes radiation, but when applied after irradiating, the reduced effect eventually disappears. It is now generally considered that the enhanced effect seen with a simultaneous or close application results from radio-sensitization by heat, while the effect seen with a sequential treatment with a long interval is the consequence of heat-induced cytotoxicity [17,40,47].

Radiation damage induced in DNA can be repaired via several different pathways, primarily non-homologous end joining and homologous recombination [60,61]. Heat can interfere with both these pathways [62], thus preventing the repair of lethal single- and double-strand breaks from taking place (Figure 1) and contributing indirectly to cell death [63,64,65,66]. Such an effect is unlikely to be tumor-specific and can explain the similar degree of sensitization seen in tumors and normal tissues. An alternative mechanism proposed to explain radio-sensitization involves an improvement in oxygen delivery to tumors (Figure 1). This is somewhat controversial. Pre-clinical studies demonstrate that mild heat temperatures (≤42 °C for 1 h) improve tumor oxygenation [13,67,68,69,70], probably via a heat-induced improvement in tumor blood flow [68,71]. Clinical studies also reported improvements in tumor oxygenation following mild temperature heating [72,73]. However, while increases in tumor blood flow and oxygenation occur during the period of heating at mild temperatures, they generally return to normal when the heating stops [13,68,71,74]. Furthermore, radio-sensitization increases with heating temperature, and although temperatures above 42 °C may cause a transient increase in blood flow and oxygenation during heating, there is a rapid decrease in blood flow immediately after heating (Figure 1), and this would be expected to substantially decrease tumor oxygenation status [68,71,74]. Finally, improved tumor oxygenation can only enhance radiation response if present at the time the radiation is applied [75]. This would certainly explain the effects in normal tissues where there is a greater effect of heat when applied before radiation than after (Figure 2). Unfortunately, hypoxia is generally not considered to influence radiation response in normal tissues. In tumors, hypoxia is a significant issue, yet here, the radio-sensitizing effect of heat appears to be the same regardless of whether the heat is applied before or after irradiating (Figure 2), suggesting a similar mechanism for both.

With regard to the heat-induced cytotoxic effect, simply killing radiation-resistant chronically hypoxic cells probably explains the reduced, yet constant enhancement seen when radiation and heat are separated by around 4 h or more. Normal tissues do not contain the similar poor microenvironmental conditions of chronic hypoxia and low pH [18], and since we see no enhancement of radiation damage by heat with a 4 h or more interval in normal tissues, it supports the concept of hypoxia and low pH playing a major role in the cell killing mechanism.

Heat can also kill tumor cells indirectly through the induction of vascular damage [17,40,47], an effect that has also been reported in patients [76,77]. These effects are clearly dependent on the tumor type and temperature applied, but generally, while higher temperatures may induce a transient rise during the heating period, immediately after heating, there is a rapid and prolonged vascular collapse. Such a collapse would deprive downstream tumor cells of essential oxygen and nutrients, resulting in the induction of necrosis. It is likely that those cells that die first are the hypoxic cells that already exist under deprived microenvironmental conditions. Removing these radioresistant hypoxic cells via this mechanism would certainly also play a role in heat enhancing the tumor response to radiation. Regardless of whether heat enhances radiation response through radio-sensitization or cytotoxicity, both mechanisms show a time–temperature relationship in that the improved effects increase as the temperature rises or the heating time is extended [17,40,47].

While heat alone has little relevance as a therapy in clinical oncology, unless unrealistic heating times (>60 min) are used with hyperthermia temperatures (heating at 40–45 °C) or high thermal ablation temperatures (heating at >45 °C) are applied, there is certainly a rationale for combining hyperthermia (heating at 40–45 °C) with radiation to improve cancer outcome. Numerous clinical trials have now shown the benefit of adding hyperthermia to conventional radiation schedules [17,40,47]. With improvements in technology, using a reduced number of fractions while using higher doses per fraction is becoming more common. However, there remains the question of how to combine hypofractionation and hyperthermia when the radiation treatment kills tumor cells directly but potentially has an extravascular mediated effect that can change the tumor microenvironment and thus profoundly affect cellular heat response. We will now critically review what is currently known about combining hypofractionation and hyperthermia pre-clinically, how this combination is being applied clinically, and the implications for future studies.

## 3. Pre-Clinical Studies with Hypofractionation and Hyperthermia

### 3.1. In Vitro Studies

There are numerous in vitro studies where a single high radiation dose and hyperthermia have been combined, as illustrated in Figure 2 left. However, there are very few in vitro studies using fractionated treatments, especially with high radiation doses, as shown in Table 1.

The few fractionated and heat in vitro studies were quite surprising, as it was shown by Henle et al. that cell-killing kinetics of heat and radiation in a fractionated regimen are more complex and cannot always be translated from single-dose studies [78]. While the results from in vitro pre-clinical studies cannot be directly translated into clinical practice, they are an excellent source for investigating issues on heat temperature, heating period, sequencing, and the time interval between radiation and heat and understanding some of the mechanisms involved. They certainly help guide relevant in vivo studies. The in vitro single-cell studies listed in Table 1 involved using primarily human breast cancer cells with heat (39–44 °C) administered either 2 h before [81,82,83] or after [83] the first radiation treatment of 2 × 5 Gy, or 2 h before the first of a 5 × 2 Gy schedule [81,82]. The general findings were as reported with single radiation treatments, namely that heat enhanced radiation response, with the effects being independent of sequence but dependent on the heat temperature with the highest temperature producing the greatest effect. One other in vitro breast study [80] applied the heat 4 h before or after the first and last radiation fraction when giving 4 × 4 Gy or 6 × 3 Gy. However, radiation schedules had little effect on necrotic cell response, but there was a significant difference in apoptotic cell death. Armour et al. used a glioblastoma model and gave five fractions of 5 Gy, but low heat temperatures were applied at various times to really investigate the effect of whole-body hyperthermia, so it is not really relevant to standard radiation/heat treatments [79].

A more recent study into the combination of hypofractionation and hyperthermia utilized multicellular organoids derived from cervix cancer patients [84]. They reported no sequence dependency but a clear effect of interval on the interaction, with a simultaneous treatment having the greatest effect that decreased as the interval increased. This was exactly the same as they found for single-dose treatments in a variety of cervix cancer cell lines and consistent with the in vitro cell studies shown in Figure 2. Overall, the limited in vitro studies with hypofractionation show nothing that contradicts studies done with large single doses but do not supply any additional relevant information that can help understand the potential role of hypofractionation and hyperthermia.

### 3.2. In Vivo Rodent Studies

Although there is a similarity in the response of cells in vitro and tumors in vivo in respect to heating time and temperature [9,12], cells in culture lack a functional vascular supply, and this could play a critical role in the overall response when combining hypofractionation and heat. Extensive in vivo studies are necessary, and those investigating hypofractionation and hyperthermia are summarized in Table 2.

Various rodent tumor models have been used, including breast, melanoma, sarcomas, cervix, prostate, hepatoma, and even retinoblastoma, with the endpoints being tumor cell survival, tumor regrowth and control, tumor necrosis and apoptosis, animal survival, and metastasis formation and even vascular changes and oxygenation effects. Radiation treatments varied anywhere from 1–25 fractions, typically separated by 24–48 h, with the dose/fraction being highly variable. The applied heating method/treatments also greatly varied, involving the use of iron oxide nanoparticles and alternating magnetic field (AMF), water bath, local focused ultrasound, radiofrequency, coaxial, and even whole-body hyperthermia. Specific details on the advantages and limitations of each of these heating methods and their relevance to clinical heating approaches have been reviewed [117]. While the temperatures ranged from 39–45 °C, the heating times were variable, with a maximum of 60 min applied either just once or with every radiation treatment. The sequences included heat applied both before and after irradiating, with the intervals ranging from a simultaneous application to up to 24 h.

Most studies concluded that the effect of radiation with heat was superior to radiation alone. However, since there was little consistency between the studies, generalizations became difficult. The heating times and temperatures varied greatly, although at least one-third did use 1 h heating, which is generally consistent with the clinical application of heat. Single-dose studies have shown that the higher the heat temperature and the longer the heating period, the greater the effect of heat when used alone or when combined with radiation [17,40]. Although only a few of the studies listed in Table 2 investigated these issues, we can assume the same is true for the fractionated schedules. Several studies investigated the effect of varying the time interval between radiation and heat, and as for single treatments, the shorter the interval, the greater the enhancement [87,88,98]. The importance of the number of heat treatments was also investigated in some studies, but here, the results were somewhat controversial. Overgaard reported that the simultaneous treatment of a C3H mouse mammary carcinoma with five fractions of radiation and heat was superior to giving five fractions but with only one simultaneous heat treatment [88]. This was not the case with the radiation and heat separated by a 4 h interval. In that sequential situation, one or five heat treatments with the five radiation fractions were identical. Similar results were seen using the sequential 4 h radiation–heat interval in the same C3H mouse mammary carcinoma when applying 3 × 15 Gy in a 1-week period [109]. In that study, heating after only the last irradiation-induced the same degree of enhancement as heating after every irradiation. Simultaneous radiation–heat treatment was not investigated. Also using a C3H mouse mammary carcinoma model but irradiated with 20 fractions in 20 or 26 days, Marino and Cividalli found that the enhancement of radiation response increased with the number of heating applied almost simultaneously [98]. However, unlike the previous studies, in the “20 radiation fractions in 20 days” schedule, using four heatings applied 4 h after irradiating was significantly superior to one heating. A C3H mammary carcinoma was used in the three studies described above, and the endpoint was always local tumor control. However, the radiation schedules were different, as were the heating temperatures, and these could have influenced the results. Those studies also only reported what happened with a simultaneous radiation–heat treatment and/or the effect giving heat 4 h after irradiating. What happens between these two extremes, which is typically the period used in the clinical setting, is largely unknown. Clearly, these variables need further investigation before making concrete recommendations for the clinic.

Despite the various heating methods applied in the studies mentioned in Table 2, only one study investigated the effect of different heating methods (water bath and ultrasound) combined with a hypofractionated regimen on tumor response. They stated that the tumor control achieved by ultrasound heating was better than water bath heating but only when hyperthermia was combined with high dose rate fractionated irradiation [92]. This was an interesting observation since they reported that water bath heating had a more uniform heating of the KHT sarcoma compared to ultrasound heating. They suggested the possibility of the non-thermal effects of ultrasound further contributing to the hyperthymia enhancement, although this could not be proven by their outcome. Thus, the hyperthermia–hypofractionated radiation interaction may be more complex, with several additional parameters possibly influencing the overall effectiveness of the combined treatment modality. Further investigations of these parameters are clearly warranted.

### 3.3. Larger Animal (Canine and Feline) Studies

There are suggestions that pre-clinical studies using canines and felines with spontaneous tumors more closely resemble the clinical situation. The genetics of dogs and cats are much closer to those of humans compared to the genetics of rodents and other models typically applied for translational research [118]. Spontaneous tumors in dogs and cats have been shown to be caused by similar exposure to environmental carcinogens compared to humans. Furthermore, the typical molecular and cell signaling pathways giving rise to these spontaneous tumors are also found in human tumors. Hence, spontaneous canine and feline tumors have very comparable morphological and molecular characteristics to those of human tumors. In particular, for canine tumors, the similarity has already been seen in osteosarcomas, oral malignant melanomas, mast cell tumors, and mammary carcinomas [118]. Some of these tumors have now been the subject of investigations with hypofractionated radiotherapy and hyperthermia and are listed in Table 3.

The majority of the studies listed in Table 3 are two-arm randomized studies comparing the complete response rate of the combination therapy arm of radiation and heat to the radiation-only arm. The heat treatments typically involved high temperatures (42–44 °C) for 30 min, although one [124] also heated at 42 °C for 60 min. The radiation treatments varied between 2.5 and 10 Gy/fraction, with the heat applied once or twice/week, either 30 minutes before/after irradiating or several hours after. A complete response indicating tumor control was the endpoint in all studies. Some studies reported no beneficial effect of adding heat to radiation therapy [125,129]. In the former study, the authors produced radiation dose-response curves for tumor control and found no significant difference in the TCD_50_ values (radiation dose-producing tumor control in 50% of animals). However, there was an indication of more local control at higher radiation doses in the radiation and hyperthermia-treated groups. The absence of additional benefits at lower radiation doses may simply have reflected the low number of animals in each treatment group. The lack of response in the latter study is not so obvious because they applied a high temperature of 44 °C for 30 min within 30 min of the first two of four weekly 9 Gy irradiations, and this should have caused some benefit. All the other studies reported a benefit, probably because heating was applied on a more regular basis (one or two heatings/week), a short radiation–heat interval was used (typically < 30 min), or a high temperature was achieved (42–44 °C). The overall conclusions from these canine studies were that better tumor control was possible with better heating uniformity [121], a higher temperature minimum [124], and more heat treatments applied [126]. Tumor volume/size was shown to negatively influence the complete tumor response rates; however, one study by Dewhirst et al. reported that larger tumors were controlled better when adjuvant hyperthermia was added to radiation [119]. Furthermore, two studies reported higher complete response when high-frequency current heating was used in comparison to microwave heating [121,123]. There were suggestions that some tumor types such as melanoma, carcinoma, and sarcomas respond better to the combined heat and radiation, although this was not consistent within all the studies listed in Table 3. Finally, one study suggested that while hypofractionated radiation and heat were superior to hypofractionation alone, it was irrelevant whether the heat followed the radiation by 30 min or 4–5 h [127], which does not agree with all of the previous pre-clinical data. This may reflect the use of a high temperature (44 °C), but the authors also suggested that better tumor control was observed with the shorter interval in tumors that were less readily controlled. The larger animal study data clearly suggest critical issues that need further pre-clinical investigation.

### 3.4. Normal Tissue Response to Hyperthermia and Hypofractionated Radiation

Cancer patients are normally treated to tolerance, so it is the normal tissue response to treatment that plays a major role in influencing the treatment parameters. Single-dose studies have demonstrated that when radiation and heat are applied simultaneously or close together, there is a similar response in both tumors and normal tissues (Figure 2) and thus no therapeutic benefit if tumors and normal tissues are heated to the same temperature. However, as the interval increases, the normal tissue response decreases quicker than that in tumors, thereby resulting in an increasing therapeutic gain. Several of the in vivo hypofractionated treatment studies listed in Table 2 did try to compare the effects in tumors with those seen in normal tissues. A simultaneous radiation–heat treatment generally resulted in an enhanced response in acutely responding normal skin that was the same or even larger than the effect found in tumors [87,88,100], similar to that reported for single treatments (Figure 2). This resulted in no therapeutic benefit and appeared to be unrelated to the radiation schedule (1–10 daily treatments) or the number of heat treatments (1–5) in those studies. However, Marino and Cavidalli, monitoring skin damage after one or four heat treatments in a 20-fraction radiation schedule actually reported a reduced enhanced skin response compared to tumors, thus giving rise to a therapeutic gain [98]. Interestingly, when the number of heat treatments increased to eight, the skin reaction was substantial and greater than that seen in tumors. The two studies that investigated skin damage when hypofractionated radiation and heat were separated by 3–4 h reported no enhanced response, which is consistent with the effects reported with single treatments (Figure 2).

Several of the larger animal studies listed in Table 3 also investigated early and/or late normal tissue responses in addition to complete tumor response, primarily around the irradiated and/or heated area. The majority of the studies that reported acute and/or late normal tissue response stated that hyperthermia did not make the radiation-induced early/late normal tissue response worse, indicating the therapeutic gain of adding adjuvant hyperthermia. One study [122] reported a reduced effect of radiation and heat in terms of early normal tissue reactions, while another reported that there was a suggestion of a higher incidence of late skin reactions in the radiation–heat group when no benefit was seen in complete response (tumor control) [129]. Finally, only Denman et al. observed significantly higher hyperthermia-mediated enhancement for both early and late normal tissue responses [127]. Interestingly, all three of these studies used a high temperature of 44 °C, which suggests that above a certain hyperthermia temperature, any general conclusions become invalid, and no consistent conclusion can be made.

A few studies focused on normal tissue reactions alone, with the endpoints being early responding skin damage in mouse legs or ears [86,89,90,91,93] but also late responding mouse ear [90,91] and leg [93] deformity. Two of these studies involved heat applied with re-irradiations following previously irradiated tissues without being heated, so the results may not reflect the use of hypofractionation and heat as first-line therapy. In all the normal tissue-only studies, the radiation–heat intervals were generally short (<1 h) and the temperatures generally high (42.5–43 °C), and it was reported that heat significantly enhanced radiation-induced early damage. The study by Baker et al. showed less heat enhancement of acute skin reaction in the fractionated scheme of 3 × 10 Gy than in the single 10 Gy irradiation, independent of sequence or heating duration [89]. Two other studies [86,93] investigated the effect of temperature and found more heat enhancement for acute damage when a higher temperature was combined with radiation. Interestingly, the study by Law et al. also showed a significantly higher heat enhancement when the higher temperature was only given before radiation [86]. The effect of heat on radiation-induced late damage was not seen even at higher temperatures (43 °C) [93]. Conversely, an enhanced effect was seen for late-responding ear damage [90], although the temperature was slightly higher (43.5 °C), and the ears had been previously irradiated.

### 3.5. Conclusions from the Pre-Clinical In Vivo and Larger Animal Studies

The overall conclusions from the pre-clinical tumor studies with hypofractionated radiation and hyperthermia generally confirm the findings from single-dose studies. Namely, radiation and heat were superior to radiation alone; the higher the temperature and the better the heating uniformity, the larger the enhancement; the sequence of hypofractionated radiation with heat was irrelevant for the benefit; and applying radiation and heat at or near the same time was superior to having a long radiation–heat interval of several hours. In addition, a simultaneous/short interval generally produced a similar enhancement in both tumors and normal tissues (and thus no therapeutic benefit), while with a long time interval of several hours, the normal tissue effect was far lower than that seen in tumors (thus resulting in a therapeutic advantage). However, this was not a universal finding. Furthermore, we have absolutely no information on what happens in tumors and normal tissues when the hypofractionated radiation and hyperthermia treatment interval is between the two extremes of a simultaneous/short interval and one of 3–4 h. For most clinical treatments, one would expect to have intervals that vary from around 30 min to just a few hours. There is also the unanswered question of how often one should apply heat. Some studies suggested the more heat treatments, the better, but this was not always the case. Moreover, would one heat treatment each week be sufficient, or should heat be applied more often? Does the number of heat treatments depend on the tumor temperature achieved? What role does thermotolerance, where tissues can become resistant to a second heating, play here? Finally, when the heat temperature becomes too high (i.e., >44 °C), there seems to be an indication that the general conclusions may become invalid, suggesting that our discussion of the various issues related to a combination of hypofractionated radiation and heat might have no relevance to thermal ablation studies where temperatures in excess of 45 °C are routinely applied.

## 4. Clinical Studies with Hypofractionation and Hyperthermia

There is good clinical evidence that combining hyperthermia with conventional radiation therapy schedules significantly improves patient outcomes for a range of different tumor types [17,40]. Could a hypofractionated regimen be combined with hyperthermia to attain better hyperthermia-induced enhancement? Numerous clinical studies have investigated the potential of hypofractionated radiotherapy combined with hyperthermia, but many of those were single-arm feasibility studies involving palliative patients [130,131,132] or patients that had already undergone, and failed, prior treatment [133,134,135], or patients who were poor candidates for other treatment modalities such as surgery or chemotherapy [136]. Even though these studies provide insights into the combination of hypofractionated radiotherapy with hyperthermia, they did not compare the radiation plus heat response with radiation alone. Consequently, we have chosen to focus only on those published, peer reviewed two-arm studies investigating the therapeutic potential of combining hypofractionation and hyperthermia over that found for radiation alone regardless of whether they were randomized or not. These latter studies are summarized in Table 4.

There are two double-arm clinical studies [138,148] that have applied conventional and hypofractionated radiotherapy regimens combined with hyperthermia. For our discussions, lesions or patients treated with only a hypofractionated radiotherapy (≥3 Gy per fraction)–hyperthermia combination are shown in Table 4. The tumor types treated in the clinical studies listed in Table 4 were highly variable, although the majority were superficial tumors, not deep-seated ones. However, with the technological improvements today making the heating of deep tumors easier, it is likely that the conclusions based on the studies included in Table 4 are relevant for deep-seated tumors, although several factors such as tumor vascularization, depth, and perfusion might also influence the effectiveness of hyperthermia in all deep-seated cancer sites. The radiation treatments were administered one to five times/week, with the doses ranging from 3 to 9 Gy per fraction. Tumor heating was achieved using a variety of methods, which did not seem to influence response. The temperatures achieved were typically ≥42 °C for 30–60 min. How often the heat was applied did not appear to be a critical factor. Interestingly, two additional phase II studies, not listed in Table 4, actually investigated the effect of hypofractionated radiation and heat with patients who had locally recurrent or metastatic malignant lesions, randomized on the basis of different heat treatments. The first involved giving heat either in two or six treatments [150], while the second involved one or two weekly heat sessions [151]. The radiation treatments in both studies were similar, being either 3.5 Gy/fraction [150] or 4 Gy/fraction [151], as were the microwave heat treatments of 43 °C for at least 30 min administered after irradiating, usually within a 30–60 min interval. No apparent difference in tumor response or complication rate was found for the different arms in each study. However, the results were complicated because the studies also included patients who received conventional 2 Gy/day irradiations in 85% [150] and 26% [151] of treatments, so the significance of the number of treatments on the hypofractionation + heat schedule is debatable.

The radiation–heat intervals for the studies in Table 4 were also variable. Most studies applied the heat close to radiation (i.e., immediately before/after irradiating or within 30 min). However, intervals of 30–60 min [148] and up to 2 [149] or 4 [138,142,143] hours were also applied. The studies using ≤2 h intervals found that tumor response to the combination of radiation and heat was superior to radiation alone generally with no increase in normal tissue complications. However, one study reported that grade 3 severe erythema was significantly higher in the thermoradiotherapy group when hyperthermia was applied simultaneously (<30 min), but this disappeared when an interval of 3–4 h was introduced [143]. Another clinical study also attributed the observed enhanced acute skin reaction to the small interval between radiation and heat [141]. In addition to the radiation and heat interval, other parameters such as higher temperature and higher dose per fraction [141] were also contributing factors for acute skin reaction (dry or moist desquamation). However, these were mostly reversible post-treatment or controlled by the active cooling of the surrounding skin [140,147].

The majority of the listed studies showed that the combination arm of radiotherapy and hyperthermia had a higher complete response rate when compared to the radiotherapy alone arm. Applying a higher dose per fraction also resulted in a higher complete response rate [137,139,140,147], and this persisted in the follow-up period ranging from 6 to 24 months [140,147] or up to 5 years [144], although surgery post-thermoradiotherapy would have contributed to the 5-year prolonged complete response rate. Berdov and Menteshashvili carried out a histological analysis that showed significantly more profound changes in tumor necrosis status for rectal tumors treated with radiation and heat, which explained their outcome of a higher complete response rate in favor of the thermoradiotherapy arm [144]. The application of a higher temperature to the combined arm was also shown to result in a more enhanced complete response rate [138,145]. Masunaga et al. performed tumor downstaging and a degradation analysis where the application of a higher average intravesical temperature (T_av_ ≥ 41.5 °C) in the thermoradiotherapy arm showed superior tumor degradation and downstaging when compared not only to the radiation-only arm but also to the thermoradiotherapy arm where a lower average intravesical temperature (T_av_ < 41.5 °C) was applied [145]. Tumor size/volume has also been shown to be a prognostic factor, where complete response in larger tumors was enhanced by adjuvant hyperthermia [137,143,148]. Perez et al. reported no overall difference in complete response for radiation and heat versus radiation alone but did find a higher response in tumors that were <3 cm in size, which they attributed to smaller tumors being easier to heat [146]. On the other hand, the lack of a complete response rate in tumors ≥ 3 cm was attributed to the presence of irregular contours in the head and neck region, resulting in poor coverage for adequate heating. Hence, this study emphasizes the importance of improving the heating protocol for more efficient heating. Overall, the findings from the listed clinical studies were generally consistent with pre-clinical studies using both single and hypofractionated treatments.

## 5. General Conclusions

Adding hyperthermia to hypofractionated radiation treatments is clearly beneficial in improving tumor response, with little or no effects on normal tissue complications, provided there is sufficient interval between the two modalities. A simultaneous or short interval could possibly result in a therapeutic benefit. As reported for single-treatment studies, combining hypofractionation and hyperthermia simultaneously or with a short interval has the greatest anti-tumor effect but induces similar effects in normal tissues resulting in no therapeutic benefit. If we can avoid hitting the normal tissues, or even reduce the dose delivered to those tissues, then a therapeutic benefit would be possible. With longer time intervals between the two modalities the tumor response decreases, but the normal tissue effects are minimal or absent, allowing for a therapeutic benefit. Most studies in this area, and this also applies to hypofractionated studies, have focused on very short or very long intervals, not the more intermediate intervals typically used clinically. For radiation, the dose and schedule used do not appear to be critical factors influencing response. However, as reported in single-dose studies, with heating, the higher the temperature the greater the radiation enhancement. Although how often the heat is applied does not seem to play a role, one must always consider the possible influence of thermotolerance. This is the phenomenon whereby heated tissue becomes resistant to a second heating and has been demonstrated in vitro [42,152] and in both tumors [47,152] and normal tissues [153] in vivo. The degree of resistance and the period during which it persists are highly temperature-dependent [47]. For example, heating a C3H mammary carcinoma at 42.5 °C for an initial 30 min period prior to a subsequent heating at the same temperature showed that as the interval increased, so did the degree of resistance. This peaked with an interval of about 8 h before slowly recovering 3 days after the start of the treatment. However, if a higher temperature of 43.5 °C was used, the peak level of resistance occurred at around 16 h, was some 50–60% higher than at 42.5 °C, and disappeared at just over 4 days after treatment start [47]. Both in vitro [154] and in vivo [155] studies have shown that thermotolerance can also influence the combination of radiation and heat, regardless of whether the two modalities are given simultaneously or sequentially [152]. For example, applying five daily fractions of radiation and heat (43.5 °C for 1 h) induces the same degree of tumor control as a single radiation and heat treatment due to thermotolerance being maximal 24 h after heating at this temperature. Increasing the number of days between each fraction enhances tumor control consistent with the decay of thermotolerance. Thus, if high temperatures are achieved, it might be prudent to restrict the heating to once each week to avoid this negative issue, whereas if only lower temperatures are obtained, then heating twice weekly could be better. One good heating each week might be considered the minimum requirement.

The concept of using a reduced number of larger radiation doses with heat has several advantages over the use of conventional irradiation and heat. Reducing the total number of treatments would clearly be more cost-effective, but it would also make the treatments more convenient for patients. Furthermore, hyperthermia will only ever be applied once or twice weekly, meaning that many of the radiation treatments in a conventional schedule would be given without heating. For hypofractionation, the number of irradiations would be reduced; thus, the percentage receiving a heat treatment would be higher, so the benefits would be more likely to be greater. Clearly, additional studies directly comparing hypofractionated radiation and hyperthermia with conventional radiation schedules and hyperthermia are needed, in both tumors and normal tissues.

The exact mechanisms accounting for the enhancement of radiation response by hyperthermia are not entirely clear [156]. Heat obviously inhibits the repair of radiation-induced DNA damage repair. It can also improve tumor oxygenation status by improving oxygen delivery or directly killing the radiation-resistant hypoxic cells. There is also the possibility of an immune-mediated response being involved [156]. To what degree these various factors play a role in the overall response is unclear. With hypofractionated radiation and hyperthermia we potentially have the same mechanisms operating, but any radiation-induced vascular damage, with subsequent changes in the tumor microenvironment, could induce additional cell-killing mechanisms, further complicating the identification of relevant factors for the enhancement. Additional studies addressing these issues are clearly warranted.

## Figures and Tables

**Figure 1 cancers-16-03916-f001:**
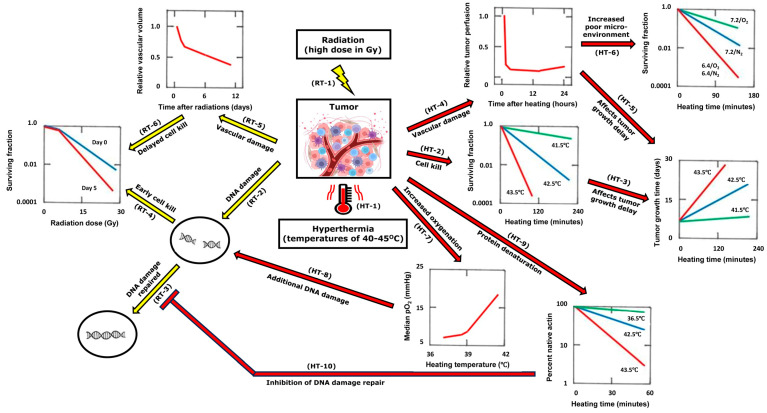
Summary of the effects of single radiation (RT) and hyperthermia (HT) treatments in tumors. When tumors are irradiated (RT-1), it causes DNA strand breaks (RT-2). Tumor cells either repair (RT-3) that damage and survive or not (RT-4), the latter resulting in cell killing when measured immediately (day 0) after irradiating. With higher doses, radiation can also damage tumor vasculature (RT-5), which can result in additional cell killing (RT-6) at later times (day 5). When tumors are heated (HT-1), there is also direct cell killing (HT-2) in a time–temperature relationship. This will influence tumor growth delay (HT-3) in a similar fashion. At higher temperatures, heat also induces vascular damage (HT-4), which can indirectly affect tumor growth delay (HT-5) and increase the poor microenvironmental conditions (low pH and hypoxia) within tumors, and cells under such conditions are more sensitive to heat-induced cell killing (HT-6). Heat also transiently increases tumor oxygenation (HT-7), which can enhance radiation-induced DNA damage (HT-8). It can also cause protein denaturation (HT-9), which, by affecting the proteins involved in DNA repair, will also give rise to additional radiation-induced cell killing (HT-10). Based on published data [9,10,11,12,13,14,15,16,17]. Tumor image adapted from biorender.

**Figure 2 cancers-16-03916-f002:**
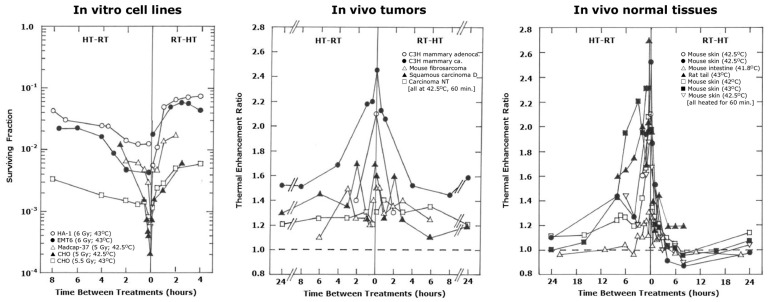
The effect of sequence and interval on the interaction between radiation and heat. In all panels, radiation (RT) was applied at time 0, with heat (HT) given at different times before (HT-RT) or after (RT-HT) irradiating. (**Left**) panel: shows in vitro survival for the various cell lines indicated on the insert following the stated radiation and heat treatments. Data redrawn from [48,49,50,51]. (**Middle**) panel: shows the level of enhancement of the radiation response by heat in tumors. The various tumor types are indicated on the insert; all tumors were heated at 42.5 °C for 60 min. Data redrawn from [52,53,54,55]. (**Right**) panel: shows the level of enhancement of the radiation response by heat in normal tissues. The different normal tissues are indicated on the insert; all were heated for 60 min at the various temperatures shown. Data redrawn from [53,54,56,57,58,59].

**Table 1 cancers-16-03916-t001:** In vitro pre-clinical studies combining hypofractionation and hyperthermia.

Cell/Tumor Type	Radiation (RT)	Hyperthermia (HT)	RT-HT Sequence/Interval	Endpoints	Conclusions	Ref.
CHO Cells	1^st^ RT = 4 Gy; 2^nd^ RT = various doses (0/2/4/8/24 h interval)	Water bath; 45 °C for 10 min (0–2 HT fx)	HT 2.5 min before 1^st^ RT and/or 2^nd^ RT	Clonogenic cell survival	Cell-killing kinetics of fractionated RT and HT are more complex and not always the same as single treatments	[78]
9L gliosarcoma	5 × 5 Gy (Within 3 days)	Water bath; 41 °C various times	8 h before 1^st^ RT to end of RT (~60 h)	Clonogenic cell survival	Effect of WBH so not relevant	[79]
MCF-7 and MDA-MB-231 breast	4 × 4 Gy or 6 × 3 Gy	HT chamber; 41 °C for 1 h	HT 4 h before or after 1^st^ and last RT	Apoptosis and necrosis	HT before RT increases MDA-MD-231 apoptosis independent of RT schemes; HT after RT increases MCF-7 necrosis slightly only for 4 × 4 Gy RT	[80]
MCF-7 and MDA-MB-231 breast	2 × 5 Gy (days 0 + 3) or 5 × 2 Gy (24 h intervals)	Warm water and microwaves; 39, 41, 44 °C for 1 h	HT-2 h-1^st^ RT	Apoptosis and necrosis	HT temperature dependent benefit, independent of RT scheme; microwaves better than warm water	[81]
MCF-7 and MDA-MB-231 breast	2 × 5 Gy (days 0 + 3) or 5 × 2 Gy (24 h intervals)	Microwaves; 39, 41, 44 °C for 1 h	HT-2 h-1^st^ RT	Apoptosis and necrosis	HT temperature-dependent benefit, independent of RT scheme	[82]
MCF-7 and MDA-MB-231 breast	2 × 5 Gy (24 h intervals)	Microwaves; 39, 41, 44 °C for 1 h	HT-2 h before/after 1^st^ RT	Apoptosis and necrosis	No sequence dependence; HT temperature-dependent benefit	[83]
Organoids from cervical cancer patients	3 × 4 Gy (24 h intervals)	Water bath; 42 °C for 1 h	HT 0–8 h before or after 1^st^ RT	Organoid number ratio from day 8 to day 1	Shorter interval had the greatest HT effect	[84]

Abbreviations: fx: fractions; Gy: gray; min: minutes; RT: radiation; h: hour; HT: hyperthermia; Ref: reference; sec: seconds; WBH: whole-body hyperthermia; ×: number of times.

**Table 2 cancers-16-03916-t002:** In vivo tumor and normal tissue pre-clinical studies combining hypofractionation and hyperthermia in rodents.

Tumor and/or Normal Tissue (NT)	Radiation (RT)	Hyperthermia (HT)	RT-HT Sequence/Interval	Endpoints	Conclusions	Ref.
Fibrosarcoma (mice)	4 × 4.5–9 Gy (1–4 day intervals)	Water bath; 4 × 43.1 °C/15 min	RT- < 2 min-HT	Ex vivo cell survival, tumor growth, mouse survival	RT+HT superior to RT alone for all endpoints	[85]
Mouse ear	Various doses in 1/2/5/10 fx (24 h interval)	Water bath; 42.5 °C/30 min or 43.5 °C/30 min	HT 6 min before/after RT (HT with all RT fx)	Acute normal skin	TER dependent on temperature; HT-RT(TER) > RT-HT(TER)	[86]
Fibrosarcoma (mice) and mouse leg	15–50 Gy in 1/2/5 fx (24 h intervals)	Water bath; 42.5 °C/1 h	RT-0/3 h-HT (HT with all RT fx)	Tumor growth and acute normal skin	TGF for 1 fx RT seen with 3 h interval; no TGF for 2/5 fx RT	[87]
C3H mammary carcinoma (mice) and surrounding skin	Various doses in 1/5 fx (24 or 72 h intervals)	Water bath; 42.5 °C/1 h	RT-0/4 h-HT (HT with all or last RT fx)	Tumor control and acute normal skin	TGF seen with RT-4 h-HT; TGF improved with 72 h interval between RT fx; TGF seen with both all/last HT	[88]
Mouse leg	1 × 10/15/20 Gy or 3 × 10 Gy (48 h intervals)	Ultrasound; 42.5–43 °C/0.5–1 h	HT ≤ 0 –1 h before or after RT	Acute normal skin	TER for 3 × 10 Gy RT < TER for 1 × 10 Gy RT; TER independent of sequence or HT duration	[89]
Mouse ear (Re-irradiation)	RT: 1 × 17 Gy or 10 × 3.4 Gy daily fx; reRT: various doses	Water bath; 43 °C/15 min	RT-3 to 12 months-reRT-6 min-HT	Acute normal skin, late ear deformity	Previous RT increases HT sensitivity; no effect of fractionation for all endpoints	[90]
Mouse ear (Re-irradiation)	RT: 1 × 19 Gy or 10 × 3.8 Gy daily fx; reRT: various doses	Water bath; 43 °C/12 min	RT-10 months-reRT; HT given 6 min before or after reRT	Acute normal skin, late ear deformity	Previous RT increases HT sensitivity; no effect of fractionation; HT-RT showed more effect than RT-HT for all endpoints	[91]
KHT sarcoma (mice)	3 × 10 Gy (2/3/4 days intervals) given with low (0.19 Gy/min) and high (2.12 Gy/min) dose rates	Water bath; 42.5 °C/30 min Ultrasound; 42.5 °C/30 min	HT-0.1 h-RT	Tumor control and metastasis	HT enhanced RT with both endpoints; high dose rate gave better HT enhancement; HT by ultrasound had better enhancement with high dose rate RT only	[92]
Mouse leg	1 × 30 Gy or 6 × 6 Gy (48 h intervals)	Water bath; 37–43 °C/45 min after all RT fx	RT-0.1 h-HT	Acute and late normal skin, carcinogenesis	HT effects seen for acute; late damage: RT+HT less than RT alone; no carcinogenic effect; no effect of fractionation	[93]
R1H rhabdomyosarcoma (rats)	25 × 3 Gy (5 fx/week)	Microwaves; 2 × 43 °C/1 h (Monday/Friday)	RT-10 min-HT	Vascular changes	RT+HT showed more vascular damage than RT alone	[94]
Breast carcinoma Tx and Sarcoma 37 (mice)	2 × 8.5 Gy (48 h interval)	No heat method stated; 2 × 43.5 °C/30 min (48 h interval)	HT after RT	Tumor necrosis	RT+HT showed more tumor necrosis than RT alone for both tumor models	[95]
Y-79 eye implanted retinoblastoma (mice)	3–9 × 3 Gy (3 fx/week)	Coaxial heating; 1–3 × 43 °C/30 min or 1–3 × 45 °C/30 min	RT-15 min-HT	Tumor control	HT enhanced 3/6 × 3 Gy; 8/9 × 3 Gy alone too effective for HT effect to be seen	[96]
R1H rhabdomyosarcoma (rats)	25 × 3 Gy (5 fx/week)	Microwaves; 2 × 43 °C/1 h (Monday/Friday)	RT-10 min-HT	Vascular changes	RT+HT showed more vascular damage than RT alone	[97]
C3H mammary carcinoma (mice) and mouse leg	Various doses in 20 fx given daily or 20 fx in 26 days (weekend gap)	Water bath; 43 °C/1 h (1 HT with 1^st^ RT only, 4 HT every 5 or 7 days, 8 HT every Monday and Thursday)	RT-0 or 4 h-HT	Tumor control and acute normal skin	4 HT fx with 0 or 4 h intervals showed the best TGF	[98]
RIF-1 rhabdomyosarcoma and surrounding skin (mice)	10 × 4 Gy (72 h interval)	Radiofrequency; 10 × 45 °C/15 min or 10 × 43 °C/60 min (72 h intervals)	RT followed by HT	Tumor regression, regrowth, curability and acute normal skin	RT+HT > RT alone for all tumor endpoints; RT+HT has similar skin reaction to RT alone;	[99]
FSa-II fibrosarcoma (mice) and mouse leg	Various doses in 5 daily fx	Water bath; 43.5 °C/45 min	HT-24 h-1^st^ RT; 1^st^ RT- ≤ 2 min-HT; 5^th^ RT- ≤ 2 min-HT; 5^th^ RT-0.5 h-HT; 5^th^ RT-4 h-HT; 5^th^ RT-24 h-HT	Tumor growth and acute normal skin	HT enhanced RT in tumors and acute NT, so no TGF for any interval or sequence.	[100]
	Various doses in 10 daily fx		HT-24 h-1^st^ RT; 1^st^ RT- ≤ 2 min-HT; 10^th^ RT- ≤ 2 min-HT; 10^th^ RT-24 h-HT	Tumor control and partial foot atrophy (late)	HT enhanced RT in tumors and late NT, so no TGF for any interval or sequence	
FSa-II fibrosarcoma and MCa mammary carcinoma (mice) and mouse leg	Many doses in 2/5/10/20 fx (24 h intervals for 2/5/10 fx; 20 fx with 6/18 h intervals)	Water bath; 43.5 °C/45 min	HT-24 h-1^st^ RT	Tumor control, pO_2_ and partial foot atrophy (late)	HT before RT did not affect pO_2_ or tumor growth; no TGF for either tumor model; MCa more HT sensitive than FSa-II	[101]
R1H rhabdomyosarcoma (rats)	8 × 4 Gy (2 fx/week)	Infrared; 8 × 43 °C/1 h	Simultaneous RT-HT	Tumor growth	RT+HT better than RT alone	[102]
R1H rhabdomyosarcoma (rats)	20 × 3 Gy (5 fx/week)	Microwaves; 8 × 43 °C/1 h (Monday/Friday)	RT-10 min-HT	Changes in pO_2_	RT+HT induced larger decrease than RT alone	[103]
Fibrosarcoma (mice)	1 × 20 Gy or 5 × 7.5 Gy (24 h intervals)	Temperature-controlled cage (WBH); 1 × 39 °C/1 h	HT-20 h-RT	Tumor growth	WBH + fx RT significantly better than RT alone for tumor growth; no WBH effect seen for 1 × 20 Gy	[104]
Morris hepatoma 3924A (rats)	10 × 2.5–4.5 Gy (5 days/week)	Radiofrequency; 4× (22 min ≥ 40 °C + 10 min ≥ 41 °C)	HT-<10 min-RT (HT Tuesday/Thursday)	Tumor growth	HT effect better with higher RT dose per fx	[105]
FSa-II fibrosarcoma (mice)	7 × 3 Gy (24 h intervals)	Water bath; 4–7 × 41.5 °C/1 h 24–48 h intervals	HT before/after RT	Tumor growth	Small benefit of daily HT+RT regardless of schedule; but on alternative days, HT before RT was best	[106]
B16F10 melanoma (mice)	2 × 5 Gy (5 Gy each given on days 8 and 9 post-inoculation)	Water bath; 2 × 41.5 °C/1 h on days 7 and 8 post-inoculation	HT followed by RT on day 8	Tumor growth	RT+HT better than RT alone	[107]
4T1 breast (mice)	2 × 10 Gy (24 h intervals)	Magnetic-induced HT by implanted thermoseeds; 2 × 41–45 °C/10 min	Simultaneous RT and HT	Tumor growth, metastasis, mouse survival	RT+HT better than RT alone for all endpoints	[108]
C3H mammary carcinoma (mice)	3 × 15 Gy (3 fx/week)	Water bath; 41.5 °C/1 h	RT-4 h-HT (HT with all or last RT fx)	Tumor control	HT significantly enhanced RT, irrespective of 1 or 3 HT fx	[109]
Various B16 melanoma cells lines (mice)	3 × 10 Gy (24 h intervals)	LOFU; temperature/time not stated	LOFU-24 h-RT	Tumor growth, metastasis, mouse survival	RT+HT better than RT for all endpoints	[110]
SiHa cervix tumors (mice)	3 × 4 Gy (24 h intervals)	Water bath; 42 °C/1 h	HT was applied to 1^st^ RT only (no interval/sequence stated)	Tumor growth, apoptosis	No HT effects seen in tumor growth, but significantly higher apoptosis in RT+HT	[111]
B16F10 melanoma (mice)	3 × 5 Gy (48 h intervals)	Tumor implanted INP + AMF; 2 × 43 °C/30 min	Sequence/interval not stated	Mouse survival	RT+HT better than RT alone	[112]
4T1 breast (mice)	3 × 8 Gy (24 h intervals)	Tumor implanted INP + AMF; 1 × 43 °C/20 min or (1 × 45 °C/5 min + 43 °C/15 min)	HT before 1^st^ RT only, but interval time not stated	Tumor growth, metastasis	HT significantly enhanced RT-induced growth delay with higher temperature; no temperature effect on metastasis	[113]
Lung cancer xenografts (mice)	2 × 5 Gy (48 h intervals)	Radiofrequency; 2 × 42 °C/30 min	HT-≤4 h-RT	Tumor growth, apoptosis	RT+HT > RT alone for all endpoints	[114]
TPSA24 prostate adenocarcinoma (mice)	2 × 10 Gy (48 h intervals)	LOFU; max. temp 45 °C no duration stated; (1.5 s at focal point)	LOFU-2 to 3 h-RT	Tumor growth, control, mouse survival	RT+HT > RT alone for all end points	[115]
B16F10 melanoma (mice)	1 × 15 Gy or 3 × 8 Gy (24 h intervals)	Tumor implanted INP+AMF; 1 × 43 °C/30 min	INP-3 h-AMF-1 h-RT	Tumor growth	RT+HT better than RT alone independent of RT scheme	[116]
SiHa cervix tumors (mice)	3 × 4 Gy (24 h intervals)	Water bath; 42 °C/1 h	HT 0/2/4/8 h before and after 1^st^ RT only	Tumor growth	Best effect with shorter interval; sequence independent	[84]

Abbreviations: AMF: alternating magnetic field; fx: fractions; Gy: gray; INP: iron oxide nanoparticles; LOFU: low-intensity focused ultrasound; min: minutes; NT: normal tissue; RT: radiation; h: hour; HT: hyperthermia; pO_2_: partial pressure of oxygen; Ref: reference; reRT: reirradiation; sec: seconds; TER: thermal enhancement ratio; TGF: therapeutic gain factor (TER tumor/TER normal tissue); WBH: whole-body hyperthermia; ×: number of times.

**Table 3 cancers-16-03916-t003:** Canine/feline studies combining hypofractionation and hyperthermia.

Tumor Type	Radiation (RT)	Hyperthermia (HT)	RT-HT Sequence/Interval	Endpoints	Conclusions	Ref.
Dogs (52) and cats (20) with various types of tumors; phase III(randomized)	4.6 Gy/fx; 2 fx/week for 4 weeks	Radiofrequency 44° C/30 min; 1x/week	HT-10 min-RT	Tumor control and early/long-term NT response	Overall CR for RT+HT > RT alone; HT only not effective; RT+HT had better CR for larger tumors; similar early reaction in all groups; no late damage	[119]
Dogs and cats (130) with various types of tumors (randomized)	4.6 Gy/fx; 2 fx/week for 4 weeks	High-frequency current or microwaves 44 °C/30 min; 1x/week	HT-10 min-RT	Tumor control	Overall CR for HT+RT > RT alone; HT prolonged response duration	[120]
					CR for RT+HT better with high-frequency current heating and more uniform heating	[121]
				Tumor control and early/long-term NT response	HT enhanced early normal tissue response, but less than in tumor; HT did not enhance late NT response	[122]
Dogs (166) and cats (70) with various types of tumors (randomized)	4.6 Gy/fx; 2 fx/week for 4 weeks	High-frequency current or microwaves 44 °C/30 min; 1x/week	HT-10 min-RT	Tumor control and early/long-term NT response	Smaller tumors, high-frequency current heating method, and higher temperature minima significantly improve CR; early/late NT response similar in RT+HT and RT alone groups	[123]
Dogs (43) with primary malignant melanoma (randomized)	4.6 Gy/fx; 2 fx/week for 4 weeks	High-frequency current or microwaves 42 °C/30 min; 1x/week 42 °C/60 min; 2x/week	HT-0 h-RT, or RT-2–3 h-HT	Tumor control	Overall CR for RT+HT > RT alone; higher control with higher temperature minima; uniform heating, smaller tumor volume, and no nodal metastasis improve CR	[124]
Dogs (38) with oral carcinomas (randomized)	2.5–5 Gy/fx; 10 fx in 22 days	Low radiofrequency current or ultrasound ≥42 °C/30 min; 2x/week	HT 3 h after RT treatments on days 1, 3, 4, 6, 7, 9, and 10	Tumor control and late NT response (necrosis)	No significant difference in the TCD_50_ value for RT alone (38 Gy) and RT+HT (33 Gy); no late NT necrosis found	[125]
Dogs (51) with various tumors phase I/II (non-randomized)	9–10 Gy/fx; 1x/week for 4 weeks	Microwaves 1 or 2x 44 °C/30 min	RT-10–20 min-HT	Tumor control	2xHT had significantly better CR than 1x HT	[126]
Dogs (113) with various tumors; phase III (randomized)	3.5 Gy/fx; 3x/week for 14 fx	Microwaves 44 °C/30 min; 1x/week	RT-30 min-HT, or RT-4–5 h-HT	Tumor control and early/late NT response	Overall CR for RT+HT > RT alone regardless of interval; HT significantly enhanced early and late NT response	[127]
Dogs (64) with spontaneous soft tissue sarcomas; phase II (randomized)	3.5–5.5 Gy/fx; 10 fx in 22 days	Ultrasound 42 °C/30 min; 2x/week	RT-3 h-HT	Tumor control and late NT response	Overall CR for RT+HT similar to RT alone but heat significantly prolonged local tumor control; late NT response similar in both arms	[128]
Dogs (145) with spontaneous head and neck tumors; phase III (randomized)	9 Gy/fx; 1x/week for 4 weeks	Microwaves 2× 44 °C/30 min;	HT within 30 min of 1^st^ and 2^nd^ RT	Tumor control and early/late NT response	Overall CR for RT+HT similar to RT alone; similar early NT toxicity in both arms; but more late NT response (skin reactions) in RT+HT group	[129]

Abbreviations: CR: complete response rate indicating tumor control; fx: fractions; Gy: gray; HT: hyperthermia; min: minutes; NT: normal tissue; RT: radiation; h: hour; Ref: reference; sec: seconds; TCD_50_: tumor control probability (radiation dose required to control 50% of tumors); ×: number of times.

**Table 4 cancers-16-03916-t004:** Clinical studies combining hypofractionation and hyperthermia.

Study Characteristics	Radiation (RT)	Hyperthermia (HT)	RT-HT Sequence/Interval	Clinical Findings	Ref.
**(a) *Non-randomized studies***					
Multiple recurrent malignant melanoma lesions (99) in 38 patients RT (54 lesions) RT+HT (45 lesions)	13 × 3.3 Gy or 10 × 4 Gy; 2x/week 7 × 5.5 Gy or 6 × 6.6 Gy; 1x/week	Radiofrequency 42–43.5 °C for 30 min; 1 or 2x/week	HT-3–6 min-RT	Overall CR: HT+RT > RT alone; no enhanced normal tissue morbidity; 1x/week CR better than 2x/week CR indicating better control with higher RT dose/fx	[137]
Various superficial lesions (163) in 77 patients; 71 lesions received 5–6 Gy/fx RT (31 lesions) RT+HT (40 lesions)	8 × 5 Gy; 2x/week	Microwaves or radiofrequency 42.5 °C for 45 min; 2x/week	RT-0 h-HT RT-4 h-HT	CR significantly better with higher HT temperature, higher RT dose/fx, and lower time interval between RT-HT; No increased skin damage at 45 °C due to skin around lesion cooling	[138]
	5 × 6 Gy; 2x/week	Microwaves or radiofrequency 45 °C for 30 min; 2x/week	RT-0 h-HT		
Metastatic malignant melanoma lesions (49) in 24 patients RT (8 lesions) RT+HT (38 lesions) HT (3 lesions)	3 × 6 Gy or 3 × 8 Gy; 1x/week 6 × 4 Gy or 6 × 5 Gy; 2x/week	Microwaves 43 °C for 1 h; 1 or 2x/week	RT- ≤ 30 min-HT	CR: RT+HT > RT alone; no CR for HT only lesions; higher Gy/fx showed better CR	[139]
Cutaneous and nodal malignant melanoma metastatic lesions (38) in 17 patients RT (17 lesions) RT+HT (21 lesions)	8 × 5 Gy; 2x/week 5 × 6 Gy; 2x/week	Radiofrequency 42.5 °C for 45 min; (5 Gy/fx); 2x/week 45 °C for 30 min; (6 Gy/fx); 2x/week	RT-0 h-HT	CR: RT+HT greater than RT alone but not significant for both schedules; CR prolonged in both arms for 6–24 months	[140]
Superficial lesions (41) in 16 patients RT (21 lesions) RT+HT (20 lesions)	6 × 4 Gy; 2 fx/week (Majority)	Radiofrequency 43 °C for 60 min; 2x/week	RT-30 min-HT	Overall response (CR+PR): RT+HT > RT alone, but CR is similar for both arms; increased skin reaction in HT group related to HT dose; also 3 late fibrosis in RT+HT arm	[141]
Superficial recurrent malignant lesions (56) in 18 patients RT (28 lesions) RT+HT (28 lesions)	10 × 3 Gy; 5 fx/week	Microwaves 41–45 °C for 45 min; 2x/week	RT-0.5–1.5 h or 3–4 h-HT (HT with 2^nd^ RT/week)	CR for RT+HT > RT alone (matched lesions); some local pain and normal tissue reactions but controlled	[142]
Cutaneous and nodal malignant melanoma metastatic lesions (115) in 36 patients RT (62 lesions) RT+HT (53 lesions)	3 × 5–10 Gy; 3 fx in 8 days	Capacitive or radiofrequency 43 °C for 60 min; 3 fx in 8 days	RT< 0.5 h-HT (simultaneous) RT-3–4 h-HT (sequential)	TER simultaneous: 1.43 (tumor) and 1.42 (skin); TER sequential: 1.24 (tumor) and 1.02 (skin); therefore, TGF sequential (1.22) greater than TGF simultaneous (1.01)	[143]
Locally advanced rectal cancer RT (59 patients) RT+HT (56 patients)	10 × 4 Gy; 3x/week	Electromagnetic 42–43 °C for 1 h; 4–5 HT fx in total	HT-10 min-RT HT started from 3^rd^ RT fx	RT+HT significantly enhanced primary tumor regression and 5-year survival	[144]
Urinary bladder cancer (49 patients); phase I/II RT (21 patients) RT+HT_low_ (12 patients) RT+HT_high_ (16 patients)	6 × 4 Gy; 3x/week	Capacitive heating 2x/week for 35–60 min; Intravesical average (T_av_) = 41.5 °C, further classified into HT_low_(<41.5 °C) and HT_high_ (≥41.5 °C)	HT immediately after RT	Tumor degradation and downstaging by thermoradiotherapy significantly higher when T_av_ ≥ 41.5 °C; local recurrence and survival similar in all three arms.	[145]
**(b) *Randomized studies***					
Various superficial tumors (237 evaluable patients); phase III RT (117 patients) RT+HT (120 patients)	8 × 4 Gy; 2x/week	Microwaves 42°C for 1 h; 2x/week	RT-0.25–0.5 h -HT	Overall CR: RT+HT similar to RT alone; significantly higher CR in smaller (<3 cm) tumors due to better heating of smaller tumors; acute/late toxicities comparable in both arms	[146]
Recurrent or metastatic malignant melanoma (128 tumors in 68 patients); phase III RT (65 tumors) RT+HT (63 tumors)	3 × 8/9 Gy; 4-day intervals	Microwave or radiofrequency 3 × 43 °C for 1 h	RT- < 30 min-HT	RT+HT had a significantly better effect on CR than RT alone, with no effect on acute or late RT reactions	[147]
Superficial localized breast cancer (56 patients in ESHO trial protocol); phase III RT (27 patients) RT+HT (29 patients)	8 × 4 Gy; 2x/week	Electromagnetic 42.5–43 °C for 30–60 min; 4–8 HT fx with ≥3 days between HT sessions	RT-0.5–1 h-HT	CR (ESHO only): RT+HT significantly greater than RT alone; odds ratio of 5.7 strongly in favor of RT+HT arm	[148]
Bone metastasis resulting in Brief pain inventory (BPI ≥ 4) (57 patients); phase III RT (28 patients) RT+HT (29 patients)	10 × 3 Gy; 5x/week	Capacitive Normal rectal (42.5 °C) or esophageal (41.5 °C) temperatures reported, as bone metastasis temperatures are not measurable 2x/week for 40 min	RT- < 2 h-HT	Compared to RT alone, RT+HT significantly increased the pain control rate and extended response duration; pain control is attributed to complete control of bone metastatic lesions	[149]

Abbreviations: CR: complete response rate indicating no presence of tumor post-treatment; fx: fractions; Gy: gray; RT: radiation; h: hour; HT: hyperthermia; PR: partial response rate indicating >50% reduction in tumors post-treatment; Ref: reference; sec: seconds; TER: thermal enhancement ratio; TGF: therapeutic gain factor (TER tumor/TER normal tissue); x: number of times.

## Data Availability

This is a review article of published studies. So, no new data were created or analyzed in this study and all studies mentioned are already in the public domain.

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
