# Peer review of "The Rationale for Combining Hypofractionated Radiation and Hyperthermia"

_cancers, 2024, doi:10.3390/cancers16233916_

Round 1

Reviewer 1 Report

Comments and Suggestions for Authors

This review article provides useful information of combination of hypofractionated radiation and hyperthermia for cancer therapy.

Comments:

1.       Schematic diagram, Figure 1. provides summary of the review paper and needed some modification for clarity. For examples: I) Fonts of X- and Y- axes labels of some of graphs are small and difficult to read. II) Both tumor growth delay (HT-3) and indirect cell kill (HT-5) result a graph displaying tumor growth time delay. The term “indirect cell kill” should be reworded something similar to growth time delay. III) It appears that HT-4, HT-6 and HT-7 represent sequential hyperthermia treatments, but it is unclear whether this hyperthermia imply to just one heat treatment or multiple heat treatments. Similar question for other sequential hyperthermia treatments, such as HT-2, HT-10 and HT-11, and others.

2.       Page 2, line 51: “indirectly in other cellular molecules, primarily water” It is unclear whether water or any other cellular macromolecules such as lipids or proteins get ionized to free radicals that diffuse to different area and cause DNA damage.

3.       Page 5 Line 172: “mild heat temperatures improve tumor oxygenation” A specific hyperthermia temperature is throughout the review paper, and it would be consistence to provide the mild heat temperatures for these studies.

4.       Page 9 Line 276, recommending providing reference: “two studies used up to 5 fractions and one even gave 10-26 fractions”. Although the first paragraph of this page is primarily a part of summary of Table 2., it is easy to track these studies with references.

Author Response

Reviewer 1

This review article provides useful information of combination of hypofractionated radiation and hyperthermia for cancer therapy

Comments 1 : Schematic diagram, Figure 1. provides summary of the review paper and needed some modification for clarity. For examples: I) Fonts of X- and Y- axes labels of some of graphs are small and difficult to read. II) Both tumor growth delay (HT-3) and indirect cell kill (HT-5) result a graph displaying tumor growth time delay. The term “indirect cell kill” should be reworded something similar to growth time delay. III) It appears that HT-4, HT-6 and HT-7 represent sequential hyperthermia treatments, but it is unclear whether this hyperthermia imply to just one heat treatment or multiple heat treatments. Similar question for other sequential hyperthermia treatments, such as HT-2, HT-10 and HT-11, and others.

Response 1: (i) We thank the reviewer for the comment and now have rectified the issue, and all figures and labels labels can be read clearly. (ii) We understand the concern here, so have changed “indirect cell kill” to “tumor growth delay”. (iii) Again, we understand the confusion, so have changed the text to clarify this. This includes stating that the effects are for single treatments, as well as removing the HT-7 arrow and related text completely. We have also now added some additional explanation for HT-2 and HT-10 in the legend, while the HT-11 arrow has been removed.

Comment 2: Page 2, line 51: “indirectly in other cellular molecules, primarily water” It is unclear whether water or any other cellular macromolecules such as lipids or proteins get ionized to free radicals that diffuse to different area and cause DNA damage.

Response 2: No, it is not macromolecules like lipids and proteins, because these could not rapidly diffuse to the DNA, so it only applies to small molecules like water and since this contributes to some 70% of mammalian cells, it is clearly the major small molecule involved. We have now modified our comments to reflect this (see page 2, lines 52-53).

Comment 3: Page 5 Line 172: “mild heat temperatures improve tumor oxygenation” A specific hyperthermia temperature is throughout the review paper, and it would be consistence to provide the mild heat temperatures for these studies.

Response 3: The treatment temperature has now been added as “(< 42OC for 1-hour)” (see page 5, line 173).

Comment 4: Page 9 Line 276, recommending providing reference: “two studies used up to 5 fractions and one even gave 10-26 fractions”. Although the first paragraph of this page is primarily a part of summary of Table 2., it is easy to track these studies with references.

Response 4: Our comments on page 9, line 276 were somewhat selective and as such were actually incorrect. We have now modified our comments to cover all the studies listed in Table 2. Thus state “radiation treatments varied anywhere from 1-25 fractions, typically separated by 24-48 hours, with the dose/fraction being highly variable” (see page 9, lines 278-279).

Reviewer 2 Report

Comments and Suggestions for Authors

The authors present the rationale for combining hypofractionated radiotherapy with hyperthermia in pre-clinical in vitro studies and in vivo rodent and larger animal studies. This is a very interest review work since it is the first one which gather all the studies for the combined hypofractionated radiotherapy/hyperthermia treatments for a wide range of cancer types.

However, I have only one major remark. The authors should survey in the literature for more recent studies on this topic. The majority of the references are old enough. There are a lot of recent original works on this subject found just with a quick look at pubmed for example. The manuscript should be thoroughly revised with recent references.

https://pubmed.ncbi.nlm.nih.gov/?term=hypofractionated+radiotherapy+and+hyperthermia&filter=datesearch.y_5.

Author Response

Reviewer 2

Comment: The authors present the rationale for combining hypofractionated radiotherapy with hyperthermia in pre-clinical in vitro studies and in vivo rodent and larger animal studies. This is a very interest review work since it is the first one which gather all the studies for the combined hypofractionated radiotherapy/hyperthermia treatments for a wide range of cancer types.However, I have only one major remark. The authors should survey in the literature for more recent studies on this topic. The majority of the references are old enough. There are a lot of recent original works on this subject found just with a quick look at pubmed for example. The manuscript should be thoroughly revised with recent references https://pubmed.ncbi.nlm.nih.gov/term=hypofractionated+radiotherapy+and+hyperthermia&filter=datesearch.y_5.

Response 1: We agree with the reviewer that any up-to-date review must include literature for recent studies. This we have tried to do, where 19 (>12%) references in our review are from the last 5-years and 36 (>36%) from the last 10 years. However, many of the studies relevant to the topic under review, were published more than 10-years ago and must be included. We followed the reviewer’s advice and entered the terms “hypofractionated+radiotherapy+hyperthermia” in PubMed and found 18 hits. Of these, only 13 could be considered recent (i.e., within the last 10 years). One of these 13 studies was a pre-clinical one (Kotter et al., 2015, Radiat. Oncol.) and is already included in table 1. The other 12 publications were clinical. However, of these, 5 did not meet the criteria for inclusion because they either focused on heat alone, or did not actually include heat with hypofractionation. The remaining 7 publications did involve hypofractionation + heat but were actually single arm studies and as we stated on page 14, such studies were considered irrelevant to our review where we compared hypofactionated radiation against hypofractionated radiation+heat

Reviewer 3 Report

Comments and Suggestions for Authors

The manuscript titled "The Rationale for Combining Hypofractionated Radiation and Hyperthermia" provides a well-articulated review on the rationale and potential effects of combining hypofractionated radiation therapy with hyperthermia. The manuscript is well-written overall; however, some sections would benefit from additional clarification and discussion.

1- The current version of Figure 1 is somewhat unclear. Please consider revising the sequence of events and visual distinctions to better convey the specific effects of radiation and hyperthermia on cancer cells. Additionally, while the manuscript notes that hyperthermia alone holds limited therapeutic value as a cancer treatment (lines 10–12), Figure 1 implies otherwise through its depiction of hyperthermia's effects. This concept is further reinforced in lines 209–210, where it is stated that “While heat alone has little relevance as a therapy in clinical oncology unless high thermal ablation temperatures (>45°C) are reached.” This statement would benefit from further clarification. Could you include citations or examples to substantiate this claim and provide a more detailed explanation of its basis?

2- The manuscript references water baths and microwave methods as hyperthermia techniques (Table 1) and briefly mentions ultrasound and magnetic hyperthermia (lines 278–283). Considering the range of available hyperthermia methods, a dedicated section discussing these techniques would be beneficial. In this section, please provide more detail on each technology, including advantages, limitations, and translational potential to clinical settings.

3- The review predominantly focuses on adult cancers. Expanding the discussion to include developmental cancers, specifically the implications of combining hypofractionated radiation and hyperthermia in pediatric patients, would enhance the manuscript’s scope. Table 2, for instance, summarizes studies on rhabdomyosarcoma, a developmental cancer, but lacks in-depth discussion. How might hypofractionated radiation and hyperthermia affect developmental biology in children? Additionally, please consider addressing potential secondary and long-term effects of this combination therapy in pediatric patients.

Author Response

Reviewer 3

The manuscript titled "The Rationale for Combining Hypofractionated Radiation and Hyperthermia" provides a well-articulated review on the rationale and potential effects of combining hypofractionated radiation therapy with hyperthermia. The manuscript is well-written overall; however, some sections would benefit from additional clarification and discussion.

Comment 1: The current version of Figure 1 is somewhat unclear. Please consider revising the sequence of events and visual distinctions to better convey the specific effects of radiation and hyperthermia on cancer cells. Additionally, while the manuscript notes that hyperthermia alone holds limited therapeutic value as a cancer treatment (lines 10–12), Figure 1 implies otherwise through its depiction of hyperthermia's effects. This concept is further reinforced in lines 209–210, where it is stated that “While heat alone has little relevance as a therapy in clinical oncology unless high thermal ablation temperatures (>45°C) are reached.” This statement would benefit from further clarification. Could you include citations or examples to substantiate this claim and provide a more detailed explanation of its basis?

Response 1: Figure 1 and its legend have been modified to hopefully clarify the effects. Regarding lines 209-210 then these have been modified to avoid confusion. It now states “While heat alone has little relevance as a therapy in clinical oncology, unless unrealistic heating times (>60 minutes) are used with hyperthermia temperatures (heating at 40-45OC) or high thermal ablation temperatures (heating at >45OC) are applied, there is certainly a rationale for combining hyperthermia with radiation to improve cancer outcome” (see page 5, lines 209-212).

Comment 2: The manuscript references water baths and microwave methods as hyperthermia techniques (Table 1) and briefly mentions ultrasound and magnetic hyperthermia (lines 282–284). Considering the range of available hyperthermia methods, a dedicated section discussing these techniques would be beneficial. In this section, please provide more detail on each technology, including advantages, limitations, and translational potential to clinical settings.

Response 2: Including a separate section on the various heating techniques available, with details on the advantages and limitations of each technology and potential translation into clinical settings, actually goes beyond the rationale for the current review. Furthermore, we believe that such details should be made by people with expertise in that area and there is already one excellent review that deals with this issue (i.e., Kok et al., 2020, Int. J. Hyperther). Rather than include such details we now refer to that relevant reference (see page 9, lines 282 - 284).

Comment 3: The review predominantly focuses on adult cancers. Expanding the discussion to include developmental cancers, specifically the implications of combining hypofractionated radiation and hyperthermia in pediatric patients, would enhance the manuscript’s scope. Table 2, for instance, summarizes studies on rhabdomyosarcoma, a developmental cancer, but lacks in-depth discussion. How might hypofractionated radiation and hyperthermia affect developmental biology in children? Additionally, please consider addressing potential secondary and long-term effects of this combination therapy in pediatric patients.

Response 3: Our review was not meant to focus on the use of hypofractionation + hyperthermia in adult tumors but rather designed to be a more general review covering all possible aspects of the interaction. That is why we included in vitro studies, rodents in vivo and larger animal studies. Rhabdomyosarcoma studies were simply included under animal studies for completeness. For the clinical part we only presented relevant published studies and all of these were from adults. We are not aware of any specific pediatric studies in which hypofractionated radiation and hyperthermia have been exclusively applied. Consequently, we believe any attempt to discuss this issue from an animal to human perspective, would be purely speculative and scientifically unsound.

Round 2

Reviewer 2 Report

Comments and Suggestions for Authors

Dear authors,

I propose this manuscript for publication since my major remark was answered satisfactorily.

Kind regrads

SE